# Using multiple qualitative methods to inform intervention development: Improving functional status measurement for older veterans in primary care settings

**Francesca M. Nicosia[1,2]\*, Kara Zamora[1,2], Anael Rizzo[1,2], Malena J. Spar[1,2], Molly Silvestrini[1,2], Rebecca T. Brown**[3,4,5,6]

**1** San Francisco VA Health Care System, San Francisco, California, United States of America, **2** Division of Geriatrics, Department of Medicine of the University of California, San Francisco, San Francisco, California, United States of America, **3** Geriatrics and Extended Care Program, Corporal Michael J. Crescenz VA Medical Center, Philadelphia, Pennsylvania, United States of America, **4** Center for Health Equity Research and Promotion, Corporal Michael J. Crescenz VA Medical Center, Philadelphia, Pennsylvania, United States of America, **5** Division of Geriatric Medicine, Perelman School of Medicine of the University of Pennsylvania, Philadelphia, Pennsylvania, United States of America, **6** Leonard Davis Institute for Health Economics, University of Pennsylvania, Philadelphia, Pennsylvania, United States of America

\* francesca.nicosia@va.gov

**Data Availability Statement:** The underlying data for this study consists of in-depth, qualitative interviews with (1) Veterans with aging-related

## Abstract

Functional status, or the ability to perform activities of daily living, is central to older adults' health and quality of life. However, health systems have been slow to incorporate routine measurement of function into patient care. We used multiple qualitative methods to develop a patient-centered, interprofessional intervention to improve measurement of functional status for older veterans in primary care settings. We conducted semi-structured interviews with patients, clinicians, and operations staff (n = 123) from 7 Veterans Health Administration (VHA) Medical Centers. Interviews focused on barriers and facilitators to measuring function. We used concepts from the Consolidated Framework for Implementation Science and sociotechnical analysis to inform rapid qualitative analyses and a hybrid deductive/inductive approach to thematic analysis. We mapped qualitative findings to intervention components. Barriers to measurement included time pressures, cumbersome electronic tools, and the perception that measurement would not be used to improve patient care. Facilitators included a strong interprofessional environment and flexible workflows. Findings informed the development of five intervention components, including (1) an interprofessional educational session; (2) routine, standardized functional status measurement among older patients; (3) annual screening by nurses using a standardized instrument and follow-up assessment by primary care providers; (4) electronic tools and templates to facilitate increased identification and improved management of functional impairment; and (5) tailored reports on functional status for clinicians and operations leaders. These findings show how qualitative methods can be used to develop interventions that are more responsive to real-world contexts, increasing the chances of successful implementation. Using a conceptually-grounded approach to intervention development has the potential to improve patient and clinician experience with measuring function in primary care.

functional impairment and their caregivers and (2) employees of the U.S. Veterans Health Administration. It is not possible to create a minimal data set with this qualitative data as this study did not obtain ethical approval or informed consent from participants to publicly share underlying qualitative data sets. Relevant excerpts from transcripts are included within the paper. The datasets generated and/or analyzed during this study are not publicly available but may be available upon request at the Center for Health Equity Research and Promotion of the U.S. Department of Veterans Affairs administrative offices, at 215-823-5817.

**Funding:** This research was supported by the Veterans Affairs (VA) Quality Enhancement Research Initiative (grant number QUE 15-283 to RTB; https://www.queri.research.va.gov/). The funding source had no role in the study design, data collection and analysis, decision to publish, or preparation of the manuscript.

**Competing interests:** The authors have declared that no competing interests exist.

## Introduction

Functional status, the ability to perform activities of daily living (ADLs) and instrumental ADLs (IADLs), is central to older adults' health and quality of life. Maintaining function is one of most important health outcomes for older adults [1–3]. Losing the ability to independently perform ADLs such as bathing and dressing is associated with higher rates of acute care utilization, nursing home admission, and death [4–7]. Understanding function informs the delivery of clinical interventions [8, 9] and helps providers assess the need for services and support [10–14], making it an essential component of patient-centered care.

Although leaders in geriatrics and healthcare policy have called for improved measurement of functional status for older adults, most U.S. health systems have been slow to incorporate routine measurement into outpatient care [15, 16]. Moreover, when functional status is measured, data are seldom collected in a way that is useful for clinical programs and research [15, 17–19].

The Veterans Health Administration (VA), the United States' largest integrated healthcare system, offers unique opportunities to improve functional status measurement for older adults. In 2009, VA implemented annual functional status screening among older veterans in primary care using an electronic tool administered during patient triage. Measurement focused on the ability to perform ADLs (bathing, dressing, transferring, toileting, eating) and IADLs (shopping, preparing food, managing medications, managing finances, doing housework, using transportation, using the telephone) [20, 21]. In these efforts, there was little stakeholder input regarding design and implementation, and barriers and facilitators to implementation were not formally assessed. An evaluation later showed that measurement uptake was low and of varying quality, partly due to the length and complexity of the screening tool; some medical centers measured function consistently and others not at all [16]. Furthermore, data were seldom used to inform patient care.

Over the past decade, VA and other health systems have increasingly incorporated patient and clinician perspectives in intervention development to inform successful implementation [22, 23]. In the present study, we developed an intervention to improve routine measurement of function in VA primary care settings informed by perspectives of patient, clinical, and operational stakeholders.

### Using qualitative methods to inform intervention development

Qualitative methods are well suited to inform the development of clinical interventions that are responsive to the needs and preferences of those who deliver and receive them [24–26]. Specifically, interviews can establish acceptability among participants and increase face validity, especially for clinical assessments using health information technology (HIT) [27–29]. However, the foregrounding of qualitative inquiry in the development of interventions remains limited; most reports of intervention development lack accounting of theories or specifics of qualitative methodologies that are used [30, 31].

Similarly, qualitative methods are key for developing effective strategies for implementing interventions and are widely used in formative evaluations to identify factors that influence intervention uptake, adaptation, implementation, sustained use, and spread [32–37]. Numerous implementation science frameworks provide concepts to guide the identification of barriers, facilitators, and contextual factors that influence implementation [38–40]. However, these frameworks seldom provide theoretical concepts to critically examine complex social dynamics within healthcare settings, particularly in the context of HIT use [41–46].

"Sociotechnical" approaches can complement implementation science concepts by increasing our understanding of the dynamic and complex nature of healthcare systems [47–51].

Sociotechnical theories posit that organizations are complex social and technical systems with recursive and iterative relationships among HIT, users, and workflows [48, 49]. Sociotechnical theories assert that HIT is not neutral; technologies are designed by individuals with inherent assumptions about intended users, how they use technologies, and the impact on clinical work. By incorporating sociotechnical and implementation science concepts during the developmental phase, interventions using new HIT have the potential to be more responsive to real-world contexts, increasing the chances of successful implementation.

This study describes the use of multiple qualitative methods, informed by concepts drawn from implementation science and sociotechnical analysis, to develop a patient-centered, interprofessional intervention to improve functional status measurement among older veterans in primary care settings.

## Methods

We conducted a developmental formative evaluation among participants from seven geographically diverse VA medical centers to assess patient, caregiver, and clinician perspectives and contextual factors and impacting functional status measurement among older adults [35, 37]. We utilized both rapid and thematic qualitative analysis to inform development of intervention components. The study team included two medical anthropologists and health services researchers (FMN, KZ), three research assistants trained in qualitative methods (AR, MJS, MCS), and a geriatrician-researcher (RTB). This study was approved by the institutional review boards of the San Francisco VA Medical Center and the University of California, San Francisco (combined approval no. 15–17697) and the institutional review board of the Corporal Michael J. Crescenz VA Medical Center (approval no. 1581262–6). All patient and caregiver participants provided written informed consent and VA employees provided verbal consent prior to enrollment in the study. Human subjects' data were collected between March 2016 and October 2016 and the study was conducted from November 2015 through November 2022. The authors had access to information that could identify individual participants during and after data collection.

### Data collection

We recruited clinicians and operational staff from six VA medical centers. We used a criterion sampling approach to sample centers with varying characteristics as follows [52]. First, we selected at least one center from each of VA's five geographic regions. Second, we selected centers based on their approach to measuring function. We identified three approaches by analyzing VA data: (1) routine use of a structured screening tool to collect information on ADLs and IADLs; (2) routine use of a structured tool to collect information on ADLs or IADLs; and (3) no standardized approach (Table 1) [52, 53]. We selected two centers from each category. At these centers, we recruited clinicians (nurses, primary care providers [PCPs], social workers) and operational stakeholders (information technology [IT] and performance measurement specialists, health system leaders) via e-mail and scheduled one-time telephone interviews with interested individuals. We also recruited IT and performance measurement specialists working at the regional and national level. We included participants from a range of roles to provide insights into varying aspects of measurement, from front-line screening to HIT design.

We also recruited primary care patients aged ≥65. To allow for home-based interviews and facilitate participation among individuals with functional impairment, we recruited patients from a single VA medical center where the investigators worked. This medical center had routine, standardized processes in place for measuring functional status in primary care, allowing us to elicit patient feedback on these existing processes. We stratified recruitment by functional

**Table 1. Site-Level characteristics.**

| Site | Stakeholder group[a] | | | | | | Clinic approach to measuring function | | |
|---|---|---|---|---|---|---|---|---|---|
| | LVN/RN (n = 24) | PCP (n = 24) | SW (n = 11) | HSL (n = 12) | IT/PM (n = 19) | PT/CG (n = 33) | Routine, complete, standardized | Routine, partial, standardized | Non-standardized, ad-hoc |
| Site 1 | 5 | 5 | 2 | 3 | 3 | 0 | X | | |
| Site 2 | 4 | 3 | 1 | 2 | 3 | 0 | X | | |
| Site 3 | 4 | 4 | 1 | 3 | 2 | 0 | | X | |
| Site 4 | 3 | 3 | 3 | 2 | 3 | 0 | | X | |
| Site 5 | 4 | 3 | 3 | 0 | 0 | 0 | | | X |
| Site 6 | 4 | 6 | 1 | 2 | 4 | 0 | | | X |
| Site 7 | 0 | 0 | 0 | 0 | 0 | 33 | X | | |
| Regional/ national | | | | | 4 | | | | |

[a]Abbreviations: LVN, licensed vocational nurse; RN, registered nurse; PCP, primary care provider; SW, social work; HSL, health systems leader; IT, information technology specialist; PM, performance measurement specialist; PT, patient; CG, caregiver.

status (reported by PCPs), race/ethnicity, and gender [52, 54]. For patients with cognitive impairment who were unable to provide informed consent but assented to participate, we obtained surrogate consent and recruited caregivers to participate.

We developed semi-structured interview guides informed by relevant Consolidated Framework for Implementation Research (CFIR) domains and sociotechnical concepts [39]. Interviews focused on perspectives on measuring function, including barriers and facilitators to using HIT to facilitate routine measurement. Interviews with patients and caregivers included a card sort activity [55, 56] and focused on ability to perform daily activities and functional status measurement in clinic settings, including preferences for screening and assessment, wording of questions, and clinician communication [35]. Interviews were audio-recorded, professionally transcribed, and uploaded into Atlas.ti (Version 9) for analysis.

## Data analysis

Four phases of analysis contributed to intervention development (Fig 1); additional methodologic details for developing process maps and analyzing PCP and patient interviews were previously published [53, 54, 57].

First, we used rapid qualitative data analysis [58–60] to create process maps. This team-based approach allows qualitative results to be analyzed concurrently with data collection to inform the development of interventions and implementation strategies. We organized data into summary templates and conducted matrix analysis, [58] which we used to create process maps illustrating approaches and workflows for measuring function at sites 1–6 (Microsoft Visio, Redmond, WA). To ensure accuracy, we solicited feedback on maps from participants at each site [53, 61].

Second, we conducted thematic analysis of PCP interviews to contextualize barriers and facilitators to routine measurement, including deductive and inductive coding [57, 62]. First, we developed and applied deductive codes using constructs from the study aims, interview guide, CFIR, and sociotechnical theories. Second, we serially reviewed transcripts and developed and applied inductive codes and identified emergent constructs and themes. Combined with rapid analysis across participant groups, these findings informed the development of a conceptual model for measuring function in primary care [57].

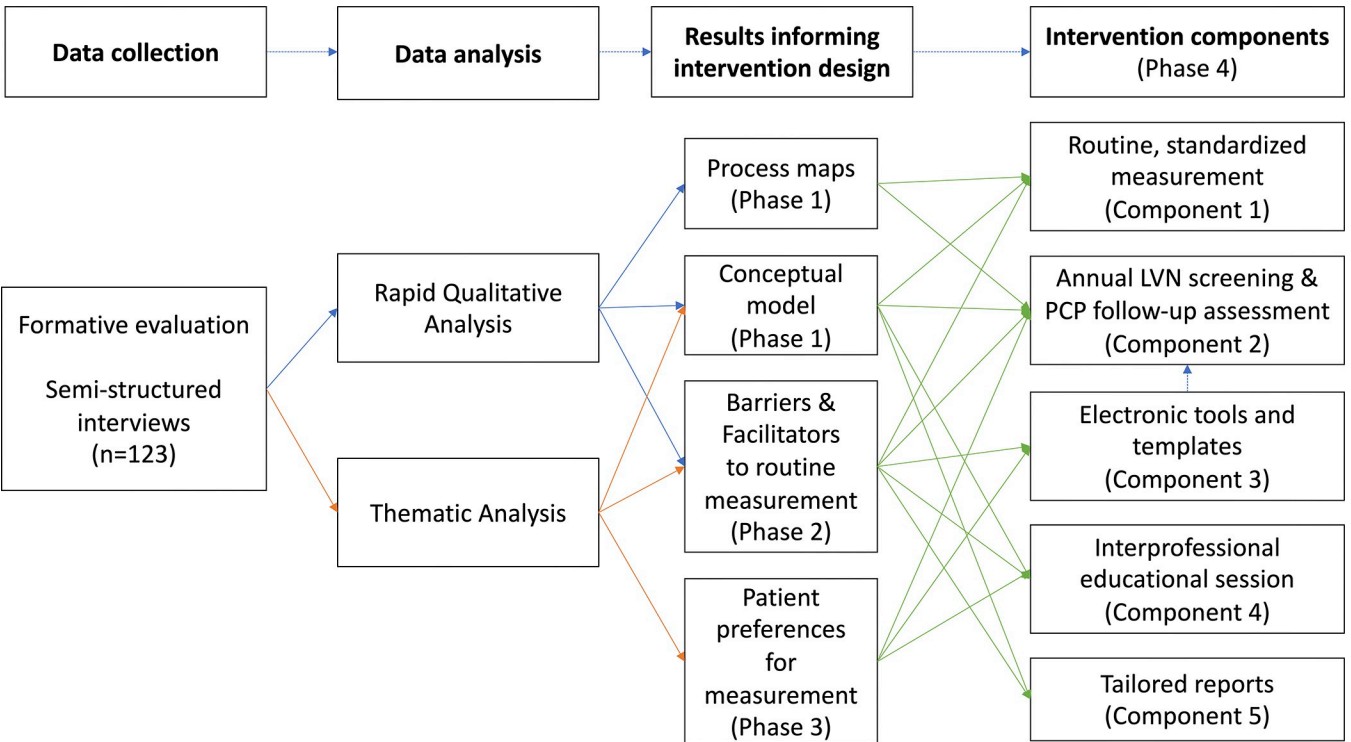

Abbreviations: LVN, licensed vocational nurse; PCP, primary care provider

**Fig 1. Flowchart of methods for intervention development.** The figure illustrates the four phases of intervention development. First, we used the findings of the rapid qualitative analyses to inform the development of process maps and a conceptual model of functional status measurement (Phase 1). Second, we used the findings of the rapid qualitative analysis and thematic analyses of interviews to identify contextual factors impacting functional status measurement (Phase 2). Third, we used findings from the thematic analyses of patient interviews to characterize patient experiences and preferences for functional assessment (Phase 3). Last, we used the findings from the first three phases of data analysis to inform the development of the five intervention components (Phase 4). Abbreviations: LVN, licensed vocational nurse; PCP, primary care provider.

Third, we used an inductive approach to qualitative thematic analysis of patient interviews to understand experience with and preferences for measurement [54, 63]. We serially reviewed transcripts and developed and applied inductive codes.

Fourth, we used the findings from rapid and thematic analyses to inform the development of intervention components (Fig 1). We used concepts from CFIR and sociotechnical analysis to map barriers, facilitators, and contextual factors identified in phases 1–3 to components, including an electronic screening tool. The VA Office of Human Factors Engineering completed Human Factors analysis of the screening tool. We then conducted usability testing and iterative refinement with intended users (licensed vocational nurses [LVNs] and PCPs) [64].

## Results

### Participant characteristics

Participants included 123 individuals from seven medical centers (Table 1). Patients' (n = 33) mean age was 81 years (SD, 10), 55% were men, 70% were white, 15% Black, 9% Latino, and 6% Asian/Pacific Islander (Table 2). More than half were independent in ADLs (58%), 21% needed help with 1–2 ADLs, and 21% needed help with ≥3 ADLs.

Clinicians included PCPs (n = 24), nurses (n = 24), and social workers (n = 11; Table 3). Of these, 80% worked in general primary care, 14% in geriatrics, and 7% in women's health.

**Table 2. Patient characteristics.**

| Characteristics | Participants (n = 33) |
|---|---|
| Age, mean years (standard deviation) | 81 (10) |
| Female, n (%) | 15 (45%) |
| Functional status (self-reported), n (%) | |
| Independent | 19 (58%) |
| Help with 1–2 ADLs[a] | 7 (21%) |
| Help with 3 or more ADLs | 7 (21%) |
| Race/ethnicity, n (%) | |
| White | 23 (70%) |
| Black | 5 (15%) |
| Latino/Latina | 3 (9%) |
| Asian/Pacific Islander | 2 (6%) |
| High school diploma, GED,[b] or less | 7 (21%) |
| Marital status | |
| Single or never married | 7 (21%) |
| Married or partnered | 6 (18%) |
| Widowed | 10 (30%) |
| Divorced | 10 (30%) |
| Housing type | |
| Private residence | 30 (91%) |
| Group home or supervised housing | 3 (9%) |
| Clinic type | |
| General primary care | 9 (27%) |
| Geriatrics primary care | 12 (36%) |
| Women's health | 12 (36%) |
| Caregiver proxy (among patients with cognitive impairment) | 5 (15%) |

[a]Activities of daily living
[b]General Educational Diploma

Operational stakeholders included health systems leaders (n = 12), IT specialists (n = 10), and performance measurement specialists (n = 9).

## Phases of intervention development

**Phase 1: Process maps and conceptual model.** Approaches to measurement varied across sites [53]. While some clinics primarily used informal approaches to assess and document function (e.g., visually assessing patients, relying on patient concerns), others used standardized electronic screening tools to assess ADLs and/or IADLs. At sites with screening tools, workflows, content, and documentation approaches differed. We developed a conceptual model illustrating the relationship between contextual factors (e.g., policies/incentives, leadership support, staffing, IT infrastructure), measurement processes (i.e., screening and assessment, documentation, use of data), and outcomes (e.g., patient access to services, improved function/quality of life) [57].

**Phase 2: Barriers, facilitators, and contextual factors.** Using findings from interviews, we identified barriers and facilitators to each aspect of measurement (screening and assessment; documentation and use of data). We also identified contextual factors and cross-cutting themes related to participants' perceptions of and preferences for measurement.

**Table 3. Characteristics of clinicians and operational stakeholders.**

| Characteristics | Participants (n = 90) |
|---|---|
| Clinicians, n (%) | |
| Primary care providers | |
| Attending physician | 17 (19%) |
| Physician fellow | 5 (6%) |
| Nurse practitioner | 2 (2%) |
| Nursing staff | |
| Registered nurse | 2 (2%) |
| Licensed vocational nurse or licensed practical nurse | 22 (24%) |
| Social worker | 11 (12%) |
| Clinic type (for primary care team members), n (%) | |
| Primary Care | 47 (80%) |
| Geriatrics | 8 (14%) |
| Women's Health | 4 (7%) |
| Operational stakeholders, n (%) | |
| Health systems leader | |
| Medical Director | 4 (4%) |
| Nurse Executive | 4 (4%) |
| Chief of Staff or Associate Chief of Staff | 4 (4%) |
| Performance measurement specialist | 9 (10%) |
| Information technology specialist | 10 (11%) |
| Years employed at VA, n (%)[a] | |
| ≤5 | 52 (60%) |
| 6–10 | 10 (11%) |
| >10 | 25 (29%) |

[a]Denominator for years employed is 87 due to missing data for three participants.

*(2a) Barriers and facilitators to screening and assessment.* Factors impacting routine screening and assessment were related to three aspects of clinic organization: workflows, communication, and infrastructure and staffing (Table 4).

Common barriers included limited time and competing clinical priorities. LVNs frequently noted how limited triage time constrained their ability to complete electronic screening tools. An LVN observed, "If the patient arrives late, then no, [the screener does not get completed]. I'll do the vitals and do the intake afterwards, because I don't want to take up the doctor's time." Given time constraints, LVNs tended to rely on ad hoc approaches to screening, such as patient observation. One LVN noted, "A lot of times I don't ask [screening questions] because most of these people are walking. . .[I]f you see them walking or talking, you already know they're competent enough to [live independently]." PCPs noted that due to time limitations during patient visits, they often focused on more pressing priorities such as acute complaints or medication management; this tendency was more pronounced in clinics without clear processes for measuring function. One PCP noted, "You're only going to be able to tackle so much. . .sometimes these patients are pretty complex and you have to really focus on the medical management and so some of those questions about function will not be asked all the time." Thus, a lack of standardized processes could result in "haphazard" approaches to measuring function, including greater reliance on patient observation and red flags.

Participants also described concerns with increased workload and "alert fatigue" from electronic screening tools. A PCP noted, "The volume of alerts I get in a given day is large and the

**Table 4. Barriers and facilitators to screening and assessment of functional status.**

| BARRIERS TO SCREENING & ASSESSMENT | ELEMENTS OF SCREENING & ASSESSMENT | | | |
|---|---|---|---|---|
| | Clinic Process: Workflows | Clinic Process: Communication | Clinic Structure and Staffing | Corresponding Intervention Components |
| Lack of time during intake (e.g., large patient load, patients late, multiple screening instruments) (LVN)[a] | X | | X | 1, 2[b] |
| Overstretched/overworked staff due to inadequate staffing and large patient load (LVN, PCP, SW) | X | X | X | 2 |
| Electronic screening tools increase workload and contribute to time burden, burnout, or "alert fatigue" (LVN, PCP, IT, PM) | X | | | 2 |
| Competing priorities<br>• Clinical care prioritized over measuring function (PCP)<br>• Completing screening tools distracts from clinical care (LVN, SW) | X | | | 1 |
| Lack of standardized clinic process/tools for measuring function<br>• Individual approaches to assessing function vary; reliance on patient observation, red flags, and/or presence of cognitive impairment (LVN, PCP, SW)<br>• Reliance on patient/family complaint to prompt measurement of function (PCP, SW) | X | X | | 1 |
| Lack of clarity around team roles and responsibilities (LVN, PCP, SW) | X | | X | 2, 4 |
| Accuracy of patient self-report varies (e.g., due to embarrassment, cognitive impairment) (PT/CG, LVN, PCP, SW, HSL) | | X | | 1 |
| Patient perception that function is not medically relevant and that assessment falls outside provider scope of care (PT) | | X | | 1, 3, 4 |
| Perception that functional status data will not be used to inform care (PCP, PT, PM) | | X | | 3, 4 |
| FACILITATORS TO SCREENING AND ASSESSMENT | Clinic Process: Workflows | Clinic Process: Communication | Clinic Structure & Staffing | Corresponding Intervention Components |
| Routine screening for functional status allows clinicians to identify impairments, understand patient needs (LVN, SW) | X | | | 1 |
| Strong interprofessional environment optimizes measurement of function (PT, LVN, PCP, SW, IT) | X | X | X | 2, 4 |
| LVN screening is a "conversation starter" that prompts patients to think about function before seeing PCP, informs provider assessment and referrals (PT, LVN, PCP, SW) | X | X | | 2 |
| Flexible workflows are necessary to accommodate varied clinic needs (e.g., screening by telephone before scheduled visits) (LVN, PCP, IT, PM)<br>• Paper forms facilitate communication and warm hand-off between LVNs and PCPs<br>• Varied forms of communication for SW consult | X | X | X | 3, 4 |
| Patients open to diverse forms of assessment (e.g., self-assessment, in-person, by telephone) (PT, LVN, SW) | X | X | | 3, 4 |
| Patients open to functional assessment if appropriate context is provided (PT, LVN, PCP, SW) | | X | | 3, 4 |
| Perception of functional status as clinically important and part of "total," "complete," or quality care (PT, LVN, PCP, SW, IT) | X | X | | 3, 4 |
| Asking about difficulty and need for help provides more complete and holistic view of patient functioning (PT) | | X | | 3, 4 |

[a]Abbreviations: PT, patient; CG, caregiver; LVN, licensed vocational nurse; PCP, primary care provider; SW, social work; IT, information technology specialist; PM, performance measurement specialist

[b]The numbered intervention components correspond to: 1. Routine, standardized functional status measurement; 2. Screening for functional status by nursing staff, follow-up assessment by providers; 3. Electronic tools and templates (LVN questionnaire; provider alert and referral menu); 4. Educational session; and 5. Tailored reports for health system leaders.

number of them that actually are meaningful are small." Inadequate staffing and large patient panels exacerbated alert fatigue, leading to "overstretched" and "overworked" staff.

Unclear interprofessional roles and responsibilities also hindered routine screening and assessment. In clinics where responsibilities for measurement were unclear, clinicians reported being less likely to measure function and feeling less motivated to do so. For example, LVNs noted that they were unsure how their screening informed PCP assessments. An LVN reported, "I often question how much [PCPs] actually look at our notes." To encourage measurement, participants emphasized the importance of "meaningful metrics," or directly connecting measurement to use of data to improve care. A PCP noted, "Does something get done with the data? Does it actually make a difference in the outcomes of the patients, or is it just another something that we're documenting and nothing's happening with it?"

Patients and clinicians frequently noted that the accuracy of a patient's self-reported function was impacted by factors including embarrassment, stigma associated with needing help with activities, and cognitive impairment. These factors could hinder screening and assessment, especially in clinics without routine measurement processes. Some functionally independent patients questioned if function, particularly IADLs, was within the scope of medical care. Other patients worried that when a clinician asked about function, they were being evaluated for nursing home admission. As one patient stated, "When I hear [those questions], my little brain is going, 'uh-oh, they're evaluating me. They're going to lock me up [in a nursing home] or something.'"

Facilitators to routine screening and assessment included standardized measurement processes. Clinicians noted that standardized screening enabled them to proactively identify impairments and understand patient needs. Another facilitator was a strong interprofessional environment. An LVN noted, "What I do like about where I work is that we function really well as a team. I think that the hand-offs are really good. So, after the [LVNs] do their thing and hand-off to the doctor or the nurse, I actually feel like we've covered the ADLs." Participants noted that LVN screening serves as a "conversation starter" that prompts patients to think about function and informs PCP assessment, showing the importance of interprofessional coordination.

Participants noted that flexible workflows were necessary to accommodate varied clinic needs. For example, nurses reported that telephone screening before scheduled visits could save valuable in-office triage time during a busy clinic session. An LVN noted, "We have a template in place that allows me to call [before the visit], and I'm not just confirming an appointment but also doing some of the [screening] that can be done [ahead of time]." Flexible workflows facilitated real-time interprofessional communication, including the use of varied strategies (e.g., paper forms, instant messaging) to alert PCPs to concerns related to patient function, and warm hand-offs between LVNs, PCPs, and social workers.

Patients' and clinicians' preferences and beliefs also facilitated measurement. Patients were generally open to varied forms of measurement, including self-assessment, in-person, or telephone screening, which supported the flexible workflows preferred by clinicians. Additionally, although some functionally independent patients perceived functional status as outside the scope of medical care, most were open to assessment when appropriate context was provided. Many patients noted that when clinicians ask about their ability to perform daily activities, it shows concern for their "wellbeing" and provides a more holistic view of their health. Similarly, most clinicians who routinely measured function perceived function as clinically important and a core element of "complete" and quality patient care.

*(2b) Barriers and facilitators to documentation and use of data.* Factors that influenced documentation and use of functional status data were related to usability of screening tools; data availability and access; and availability and knowledge of services to address impairments (Table 5).

**Table 5. Barriers and facilitators to documenting and using functional status data to improve care and outcomes.**

| | DOCUMENTATION & USE OF DATA | | | |
|---|---|---|---|---|
| **BARRIERS TO DOCUMENTATION & USE OF DATA** | **Usability of Screening Tools** | **Availability of and Access to Data** | **Availability and Knowledge of Services** | **Corresponding Intervention Components** |
| Cumbersome screening tools with poor usability and integration in clinical workflows (LVN, PCP, SW, IT)[a] | X | | | 3[b] |
| Limited response options on screening tools (LVN) | X | | | 3 |
| Limited utility of a functional status score (numeric) for informing clinical care (PCP, SW) | X | | | 3 |
| Lack of standardized data collection prevents use of functional status data, ability to track function over time (LVN, PCP, SW, HSL, IT, PM) | | X | | 1, 3 |
| Lack of population-level data about older adults limits strategic planning (PCP, HSL, IT) | | X | | 5 |
| Clinician lack of knowledge of services prevents use of functional status data to make appropriate referrals (LVN, PCP, SW) | | | X | 3, 4 |
| Lack of team coordination and communication prevents referrals to address functional limitations (LVN, PCP, SW, PM) | | | | 1, 4 |
| Access to services and supports to address functional impairments limited by geography, transportation, program funding (SW, HSL) | | | X | N/A |
| Patient eligibility issues limit access to services and supports to address functional impairments (SW, HSL) | | | X | N/A |
| Bureaucracy and red tape impede access to services and supports to address functional impairments (LVN, SW, HSL) | | | X | N/A |
| **FACILITATORS TO DOCUMENTATION & USE OF DATA** | **Usability of Screening Tools** | **Availability of and Access to Data** | **Availability and Knowledge of Services** | **Corresponding Intervention Components** |
| Standardized, user-friendly electronic screening tools and templates promote uniform documentation and care delivery (LVN, PCP, IT, PM) | X | X | | 3 |
| Numeric functional status score facilitates tracking function over time, informs efforts to improve population health (LVN, SW, IT) | X | X | | 3 |
| Free text box in screening templates captures richer information about function, improves quality of data collected (LVN, PCP, SW) | X | | | 3 |
| Integrated referral options in EHR linked to results of screening tool facilitate use of data (PCP, SW, IT, PM) | X | | X | 3 |
| Knowledge of internal and community resources facilitates use of functional status data to improve care (PCP, SW) | | | X | 3, 4 |
| Access to population-level data about older adults can inform strategic planning (PCP, HSL, IT, PM) | | X | | 5 |
| Investing in approaches to keep people functional and healthy at home is more cost-effective than institutionalization (HSL) | | | | 5 |

[a]Abbreviations: PT, patient; CG, caregiver; LVN, licensed vocational nurse; PCP, primary care provider; SW, social work; IT, information technology specialist; PM, performance measurement specialist; EHR, electronic health record

[b]The numbered intervention components correspond to: 1. Routine, standardized functional status measurement; 2. Screening for functional status by nursing staff and follow-up assessment by providers; 3. Electronic tools and templates (LVN questionnaire; provider alert and referral menu); 4. Educational session; and 5. Tailored reports for health system leaders.

At sites with existing tools for measuring function, participants pointed to poor usability as a key barrier to documentation. An LVN noted, "Now you have to click an extra button to open the [tool] and then click more buttons to answer the question, so that's kind of annoying. [The] more mouse clicks and the more steps there are, the longer it takes and we're already crunched for time as it is because we share rooms with other nurses."

LVNs also pointed to limited response options within screening tools as a barrier. An LVN observed, "I do like to have the option of a free text box at the end of each of the questions, just in case you need to add in a comment or the patient says something that you can't put in the [check box]." Some clinicians commented on the limitations of a numeric score for capturing function. A social worker noted, "The [tool] is so undescriptive. It's basically a yes or no and then you just add the points, but it doesn't tell me exactly what it means or [if] they need help with bathing." Conversely, several clinicians noted that a numeric score had value for tracking changes in function and in efforts to improve population health. A social worker reflected, "The one thing I actually like about the point value system is that you can see differences [in function over time]. . .a lot of times [functional decline] could just be a temporary thing and they get better, or they're definitely bed-bound and nothing's going to change".

An additional barrier to using data was the lack of standardized EHR data collection and documentation. Without standardized data collection, PCPs were unable to use data to inform care or track function over time. A PCP noted, "In an outpatient [setting], there's no way to go into the chart and [see] what's happening with ADLs over the last year." Social workers noted the need to "dig" through notes to glean information on function, and the lack of detail when information was available. A social worker observed, "It would be nice if there was a specific functional status note title or something with that information that could easily be found. . ., rather than looking at every person's individual notes to [find] it." Another social worker noted that when information was available, "It's just really quick–'independent.' It's not detailed".

Several factors limited clinicians' ability to use functional status data to make referrals to address impairments. These included a lack of interprofessional coordination and communication, an issue which was exacerbated in busy clinics without clear measurement processes. Additionally, clinicians often lacked knowledge of services to address impairments. A PCP said, "I honestly don't always know. . .what available services are out there. . .Things are always changing about what services [VA is] going to provide and fund".

Participants pointed to structural barriers in accessing services, including geography (e.g., limited resources in rural areas), lack of transportation, and limited program funding. A social worker noted, "Our [contracted home health] agencies are not available in certain areas, so even though [patients are] approved for [services], they can't find any agency that has staff in the area." Clinicians noted that the referral process itself could pose a barrier, due to bureaucracy and eligibility criteria. These limitations impacted patient care and outcomes. A social worker observed, "Sometimes veterans fail at home because we can only offer a few [home health aide] hours for them. . .And so, there's. . .a gap where if we had more [in-home care], it's possible that. . .[the patient] would never have to go to a skilled nursing facility".

Facilitators for documenting and using functional status data included standardized, user-friendly electronic screening tools. An IT specialist observed that usability testing was key to ensure that tools "match what the user's thinking and how they're processing the information and where to find it." A performance measurement specialist noted the importance of standardized tools for ensuring quality care, by "help[ing] keep things consistent, so that everybody answers the same way." Participants also advocated for integrating referral options within the EHR to facilitate use of data. Such integration linked measurement to referrals with the potential to improve patient outcomes, contributing to "meaningful measurement." Participants noted that knowledge of services by clinicians could help facilitate referrals, by identifying services to meet needs.

Participants pointed to broader benefits of documenting and using functional status data, including informing strategic planning and keeping patients at home rather than in more costly institutional settings. Many noted that "population data" could be used to create reports

to help better manage patients' care. Reports could help achieve the ultimate goal, summarized by one leader, of allowing "every veteran, regardless of age, to be functional and outside of an institution for as long as they can be."

Patients shared several preferences for measurement. First, most preferred being asked questions about function in person versus filling out questionnaires on their own. They described face-to-face assessment as more "intimate" and able to prompt self-reflection on one's difficulty with ADLs. In addition, patients and caregivers noted that when clinicians ask about both difficulty and need for help with daily tasks, this provides a more complete and holistic view of function. A caregiver of an 82-year-old man who needed help with ADLs stated, "I think they're two different questions. I would prefer that they were separate. Does he have trouble? Does he require help?"

**Phase 4: Intervention components.** Qualitative findings informed the development of five intervention components designed to address barriers, facilitators, and preferences for measurement (Table 6). Components included: (1) routine, standardized functional status measurement; (2) screening by nurses and follow-up PCP assessment when impairments were identified; (3) electronic tools and templates to facilitate screening and assessment; (4) an interprofessional educational session; and (5) tailored reports on functional status.

*Component 1*: *Routine, standardized measurement.* Annual, standardized measurement addressed barriers to screening and assessment, including competing priorities and a lack of

**Table 6. Intervention components and rationale.**

| **1. Routine, standardized functional status measurement** |
| --- |
| • **Component 1:** Annual measurement with a standardized tool<br>• **Rationale:** Addresses barriers to screening and assessment (competing priorities, a lack of standardized processes) and to documenting and using data (lack of standardized data location, poor team coordination); contributes to "meaningful measurement" by allowing use of standardized data to improve care |
| **2. Licensed vocational nurse (LVN) screening and follow-up PCP assessment** |
| • **Component 2:** Annual LVN screening during patient triage and follow-up PCP assessment and referral(s)<br>• **Rationale:** Team-based approach clarifies team roles and responsibilities and fosters interprofessional environment |
| **3. Electronic tools and templates to facilitate LVN screening, PCP assessment, and documentation** |
| • **Component 3a:** Validated <u>LVN screening tool</u>: (1) brief two-question pre-screener asking about difficulty performing ADLs and IADLs, [65] and, among patients who report difficulty on pre-screener, (2) in-depth screener asking about difficulty and needing help with each ADL/IADL [69–72]. Results used to auto-populate nursing note.<br>• **Rationale:** Two-part screener intended to quickly identify patients with impairment who would benefit from in-depth screening while "screening out" individuals without impairment, reducing LVN screening burden<br><br>• **Component 3b:** <u>PCP alert and referral menu</u>: If patient screens positive (i.e., reports difficulty/needing help with ≥1 ADL/IADL), PCP receives electronic alert linked to referral menu. Alert prompts PCP to review LVN screening results and perform additional assessment as needed. PCP can select up to 4 referrals: physical therapy, occupational therapy, social work, geriatric medicine.<br>• **Rationale:** Addresses stakeholder requests for integration of functional assessment into existing workflows and need for a standardized section to retrieve data on function; alert supports interprofessional approach to measurement; integrated EHR referrals address concerns that data will not be used to inform care and that clinicians lack knowledge of resources, making desired outcome (appropriate referral) more salient for PCPs |
| **4. Interprofessional educational session in VA's Training Management System (TMS)** |
| • **Component 4:** Brief pre-recorded educational session in TMS administered at beginning of implementation<br>• **Rationale:** Session reviews evidence for importance of measuring function, introduces core intervention components, and reviews patient perspectives on measurement and role of interprofessional communication |
| **5. Tailored reports** |
| • **Component 5:** Using Health Factors data, automated reports can be pulled at level of medical center, clinic, PACT, and/or individual clinician to report varying statistics (e.g., proportion of Veterans needing help with ADLs)<br>• **Rationale:** Reports provide access to population-level data to inform strategic planning and efforts to keep patients functional at home rather than in more costly institutional care |

standardized processes. This approach also leveraged facilitators, including standardized screening to systematically identify impairments and understand patient needs. Similarly, standardized measurement addressed barriers to documenting and using data, including the lack of a standardized data location and poor team coordination. Standardized measurement also contributed to "meaningful measurement" by allowing use of data to improve care.

*Component 2*: *Annual LVN screening and follow-up PCP assessment*. To operationalize this standardized process, the intervention incorporated annual LVN screening during triage and follow-up PCP assessment when impairments were identified. This team-based approach addressed a key barrier to screening and assessment, unclear team roles and responsibilities. It also leveraged facilitators by fostering a strong interprofessional environment to optimize measurement.

*Component 3*: *Electronic tools and templates*. To facilitate standardized data collection and EHR integration, we designed an electronic decision-support tool to be completed annually among patients aged ≥75. The tool had two parts: (1) an LVN screening questionnaire, and (2) a PCP alert when patients screened positive for impairment, linked to an optional referral menu.

*LVN questionnaire*. The questionnaire included suggested introductory text to read to the patient explaining the purpose of screening, followed by a two-part screener. This text provided context for measurement, proactively addressing patient concerns that asking about function was irrelevant or being used to screen for nursing home admission. It also built on observations that when context is provided, patients are more open to screening and likely to perceive it as part of quality, holistic care.

The two-part screening tool included (1) a brief two-question pre-screener asking about difficulty performing ADLs and IADLs, and (2) among patients who reported difficulty, an in-depth screener asking about difficulty and needing help with each task. This two-part structure was intended to quickly identify patients with impairment who would benefit from in-depth screening while "screening out" individuals without impairment in the most commonly-affected ADLs and IADLs. The pre-screening step reduced screening burden for LVNs, addressing concerns about limited time, increased workload, and "alert fatigue." This approach also freed time for detailed screening among patients with identified impairments. Because cumbersome tools posed a barrier to measurement, we incorporated Human Factors analysis and usability testing and iterative refinement.

We reviewed the literature to identify validated screening instruments for the two-part tool. For the pre-screener, we sought a brief instrument for identifying patients with early impairments (i.e., difficulty performing tasks, rather than needing help). Among more than 20 instruments, the American Community Survey included two items meeting these criteria, one inquiring about difficulty with ADLs and one about difficulty with IADLs [65]. We adapted these questions to include impairments which are most common in older adults and typically develop first (i.e., bathing and dressing; difficulty shopping and preparing meals) [66–68]. For the in-depth screener, we identified two candidate instruments that asked about difficulty and needing help with ADLs and IADLs: the Yale Precipitating Events Project (PEP) Instrument [69–72] and the Health and Retirement Study instrument [73]. We chose the PEP because it employs conversational language and asks about 6 ADLs and 7 IADLs.

We incorporated these measures into the screening tool design. If a patient answered no to both pre-screening questions, the EHR generated a nursing note stating that the patient screened negative for impairment. If a patient reported difficulty with bathing or dressing, an additional dialogue opened, prompting the LVN to ask questions about difficulty and needing help with each ADL; similarly, if a patient reported difficulty with shopping or preparing meals, a dialogue prompted questions about IADLs. We used a validated algorithm to score

each task as 0 (independent), 1 (has difficulty), or 2 (needs help) [69–72]. The results were used to auto-populate a nursing note detailing the patient's ability to perform each task.

Because LVNs noted that a free text box could capture richer information about function, the tool also included an optional free text section to enter notes about function or contextual factors. The tool employed conversational language to be usable either in-person or via telephone, addressing user preferences for adaptable tools.

*PCP alert and referral menu*. If a patient screened positive for functional impairment, defined as reporting difficulty or needing help with at least one ADL or IADL, the PCP received an electronic alert linked to a referral menu. The alert prompted the PCP to review the LVN screening and perform additional assessment as needed, supporting an interprofessional approach to measurement. The PCP then had the option to select up to four referrals: physical therapy, occupational therapy, social work, or geriatric medicine. Integrated EHR referrals were intended to address concerns that data would not be used to inform care, while building on perceptions that integrated referrals facilitate use of data. By providing suggested referrals, we also sought to address concerns that a lack of knowledge of services prevented using data to improve care.

*Component 4*: *Interprofessional educational session*. We developed an educational session to be completed before implementing the intervention. The purpose was to enhance clinician knowledge of the importance of measuring function, a facilitator for measurement, and introduce the intervention components. We began with an overview of the prognostic importance of functional status and how it can be used to inform clinical care. Next, we reviewed the intervention components. We reviewed patient perspectives on measurement and explained that the screening tool incorporated introductory text to address patient concerns that function is not medically relevant. We reviewed the role that interprofessional communication can play in optimizing measurement and how the intervention supports communication.

*Component 5*: *Tailored reports*. We incorporated tailored reports of population-level metrics for functional status. Reports could be pulled at the level of the medical center, clinic (e.g., medical practice, geriatrics), and provider, and tailored to report varying statistics (e.g., proportion of veterans needing help with ADLs). Reports were intended to address observations that access to population-level data informs strategic planning and efforts to keep patients functional at home rather than in more costly institutional care.

## Discussion

We used multiple qualitative methods informed by implementation science and sociotechnical concepts to develop a patient-centered, interprofessional intervention to improve measurement of functional status for older adults. Our findings suggest that using a conceptually grounded approach to intervention development has the potential to improve patient and clinician experience with measuring function in primary care.

Prior studies have not focused specifically on implementing functional status measurement in primary care. However, our study confirms and extends research examining factors that impact implementation of other types of primary care screening, including for substance use, cancer, and fall risk [74–79]. Common barriers to screening include time pressures, "alert fatigue," limited interprofessional communication, competing priorities, and limited resources to address issues identified via screening, which contribute to challenges in connecting measurement to use of data [74–79]. These similar barriers emphasize the perennial challenges of conducting routine screening in time-strapped primary care settings and the need to address these challenges when developing screening interventions.

Our findings are also consistent with studies examining barriers and facilitators to implementing HIT in VA and community primary care [80–82]. Common barriers to using HIT include overworked clinicians and poor usability; facilitators include integrating tools into clinical workflows. Our findings also provide insight into the impact of cumbersome electronic tools on clinician experience. Clinicians frequently noted the "frustration" of using such tools and how this exacerbates contextual issues such as shared clinical spaces and overworked teams. These insights illustrate the importance of using qualitative methods to understand how clinicians' experiences with HIT contribute to the U.S. epidemic of clinician burnout [83–85]. Furthermore, our findings support studies showing how electronic information transfer differs qualitatively from verbal communication (e.g., warm hand-offs) [48, 86–88].

This study has several limitations. Participants may have been more likely to enroll if they believed assessing function was important. We recruited patients and caregivers from a single VA medical center where the investigators worked; while this approach allowed for home-based interviews and facilitated participation among individuals with functional impairment, the findings for patients and caregivers may be less representative than those of participants recruited from 6 sites nationally. Our findings may not be generalizable to non-VA settings; the VA has generally longer appointment times, a predominantly male patient population, and extensive implementation of interdisciplinary primary care teams [89]. However, the identified barriers and facilitators are consistent with research in community settings [76, 78, 79, 81, 82, 87].

These dual challenges–of implementing routine screening in primary care and developing HIT that enhances rather than detracts from clinician and patient experience–underline the importance of sociotechnical frameworks for informing intervention design. Our study is part of a growing literature employing these concepts to inform intervention development and HIT implementation in healthcare settings [50, 51, 88, 90, 91]. This and prior studies show how understanding clinician experience and organizational context provides insight into unintended consequences of HIT implementation, including "alert fatigue," cumbersome tools that exacerbate time limitations, and concerns that documentation does not inform patient care. We designed the electronic screening tool to anticipate unintended consequences and increase acceptability and integration, in contrast to technology that is introduced without understanding its interaction with clinical workflows and context [48, 86, 87]. As part of this approach, we built in flexibility for how sites could implement intervention components. We also used patient findings to shape the language in the tool, to increase acceptability and face validity.

## Conclusion

We used diverse qualitative methods informed by implementation science and sociotechnical concepts to develop a patient-centered, interprofessional intervention to improve measurement of functional status for older veterans in primary care. Using a conceptually grounded approach to qualitatively-informed intervention development has the potential to improve patient and clinician experience with measuring function.

## Acknowledgments

The authors do not have any additional contributors to report.

## Author Contributions

**Conceptualization:** Francesca M. Nicosia, Malena J. Spar, Rebecca T. Brown.

**Data curation:** Kara Zamora, Anael Rizzo, Molly Silvestrini, Rebecca T. Brown.

**Formal analysis:** Francesca M. Nicosia, Kara Zamora, Anael Rizzo, Malena J. Spar, Molly Silvestrini, Rebecca T. Brown.

**Funding acquisition:** Rebecca T. Brown.

**Investigation:** Francesca M. Nicosia, Kara Zamora, Anael Rizzo, Malena J. Spar, Molly Silvestrini, Rebecca T. Brown.

**Methodology:** Francesca M. Nicosia, Kara Zamora.

**Project administration:** Francesca M. Nicosia.

**Supervision:** Francesca M. Nicosia.

**Validation:** Kara Zamora, Anael Rizzo, Rebecca T. Brown.

**Visualization:** Rebecca T. Brown.

**Writing – original draft:** Francesca M. Nicosia, Rebecca T. Brown.

**Writing – review & editing:** Francesca M. Nicosia, Kara Zamora, Anael Rizzo, Malena J. Spar, Molly Silvestrini, Rebecca T. Brown.

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
