## [Decision Letter · Decision Letter 0]

1 Aug 2023

PONE-D-23-09223Using Multiple Qualitative Methods to Inform Intervention Development: Improving Functional Status Measurement for Older Veterans in Primary Care SettingsPLOS ONE

Dear Dr. Brown,

Thank you for submitting your manuscript to PLOS ONE. After careful consideration, we feel that it has merit but does not fully meet PLOS ONE’s publication criteria as it currently stands. Therefore, we invite you to submit a revised version of the manuscript that addresses the points raised during the review process.

We look forward to receiving your revised manuscript.

Kind regards,

Mobolanle Balogun

Academic Editor

PLOS ONE

Reviewers' comments:

Reviewer's Responses to Questions

**Comments to the Author**

1. Is the manuscript technically sound, and do the data support the conclusions?

Reviewer #1: Yes

Reviewer #2: Yes

2. Has the statistical analysis been performed appropriately and rigorously? 

Reviewer #1: N/A

Reviewer #2: N/A

3. Have the authors made all data underlying the findings in their manuscript fully available?

Reviewer #1: No

Reviewer #2: Yes

4. Is the manuscript presented in an intelligible fashion and written in standard English?

Reviewer #1: Yes

Reviewer #2: Yes

5. Review Comments to the Author

Reviewer #1: Overall, the paper is well written and focuses on developing an intervention for effective assessment of functional status for older veterans. The intervention was the outcome of multiple qualitative methods of different stakeholders.

Strengths- This is an important work with a clear focus and the qualitative approach allows for better exploration.

Weakness- For a study that was set out to develop interventions, a more rigorous qualitative analysis may be expected beyond a rapid analysis.

Two concerns I have are:

1. The end-users (i.e. the patients and the caregivers) were only recruited from site 7 (Table 1). As critical stakeholders, the selection of end-users from only one site may limit the generalizability of feedback. The authors mentioned that the 7 sites were diverse- it would be helpful for a global readership to understand better why this selection for the end-users or include as a limitation.

2. Secondly, it would be important to state if the interventions developed were co-designed or not. If they were, more details would be helpful on other stakeholders who contributed, and if not, the implications of this.

Thanks

Reviewer #2: This article effectively utilizes qualitative methods to investigate the implemntation of new measures designed to ensure the full scale of functional status for older veterans is accurately captured and applied in health care settings. Data is well grounded in semi-structured interviews across multiple VA geographical regions. This article effectively utilizes matrix analysis to locate central themes and reveal compelling findings about the screening and assessment of older adults.

6. PLOS authors have the option to publish the peer review history of their article (what does this mean?). If published, this will include your full peer review and any attached files.

Reviewer #1: **Yes: **Adedoyin Ogunyemi

Reviewer #2: No

---

## [Author Response · Author response to Decision Letter 0]

9 Aug 2023

August 4, 2023

Mobolanle Balogun, MBBS, MPH, FMCPH, FWACP

Academic Editor

PLOS ONE

Dear Dr. Balogun,

We appreciate the opportunity to address the Editors’ and Reviewers’ comments and revise our manuscript. Below, please find item-by-item responses to the Reviewers’ comments, which are included verbatim. All page and paragraph numbers refer to locations in the revised manuscript. Attached please find the revised manuscript. We have tracked all changes to indicate the revised portions.

Responses to Journal Requirements:

a. Response: We have formatted the manuscript to meet PLOS ONE’s style and naming requirements.

Important: If there are ethical or legal restrictions to sharing your data publicly, please 

explain these restrictions in detail. Please see our guidelines for more information on what we consider unacceptable restrictions to publicly sharing data: http://journals.plos.org/plosone/s/data-availability#loc-unacceptable-data-access-restrictions. Note that it is not acceptable for the authors to be the sole named individuals responsible for ensuring data access.

a. Response: The underlying data for this study consists of in-depth, qualitive interviews with (1) Veterans with aging-related functional impairment and their caregivers and (2) employees of the U.S. Veterans Health Administration. It is not possible to create a minimal data set with this qualitative data as this study did not obtain ethical approval or informed consent from participants to publicly share underlying qualitative data sets. Relevant excerpts from transcripts are included within the paper. The datasets generated and/or analyzed during this study are not publicly available but may be available upon request at the Center for Health Equity Research and Promotion of the U.S. Department of Veterans Affairs administrative offices, at 215-823-5817.

a. Response: We have reviewed the reference list to ensure that it is correct and complete. We do not include any retracted articles.

Responses to Reviewer #1:

1. Overall, the paper is well written and focuses on developing an intervention for effective assessment of functional status for older veterans. The intervention was the outcome of multiple qualitative methods of different stakeholders. Strengths- This is an important work with a clear focus and the qualitative approach allows for better exploration. Weakness- For a study that was set out to develop interventions, a more rigorous qualitative analysis may be expected beyond a rapid analysis.

a. Response: We thank the reviewers for these comments. We used multiple approaches to qualitative analysis, including both rapid analysis and thematic analysis. Rapid analysis is increasingly utilized in health services research contexts (e.g., Averill, 2002, Hamilton, 2013, Nevedal et al, 2021). Our use of a matrix approach to team-based rapid analysis allowed us to analyze interviews concurrently with data collection to inform development of process maps and quickly obtain stakeholder feedback. Matrix analysis, which has been found to be as effective and rigorous as traditional deductive analysis using the Consolidated Framework for Implementation Research (Nevedal et al, 2021), was an appropriate approach to meet our study objectives. We also used thematic analysis of primary care provider interviews to contextualize barriers and facilitators to routine measurement, including deductive and inductive coding. In addition, we used an inductive approach to qualitative thematic analysis of patient interviews to understand experience with and preferences for measurement. For further description of the use of these multiple qualitative methods, see Abstract, page 5; Methods, page 6, paragraph 1; Fig. 1 Flowchart of Methods for Intervention Development; and Methods, pages 9-10.

2. Two concerns I have are:

1. The end-users (i.e. the patients and the caregivers) were only recruited from site 7 (Table 1). As critical stakeholders, the selection of end-users from only one site may limit the generalizability of feedback. The authors mentioned that the 7 sites were diverse- it would be helpful for a global readership to understand better why this selection for the end-users or include as a limitation.

2. Secondly, it would be important to state if the interventions developed were co-designed or not. If they were, more details would be helpful on other stakeholders who contributed, and if not, the implications of this.

a. Response: We thank the Reviewer for these comments. We recruited the patients and caregivers from a single site where the investigators worked in order to allow for home-based interviews and to facilitate participation among patients with functional impairment. Importantly, this site had a routine, standardized process for measuring functional status which enabled patients and caregivers to discuss their experience and provide feedback on functional status measurement in the clinic. We agree that this may limit the generalizability of the findings from this stakeholder group and have added this as a limitation as follows:

Discussion, page 28, paragraph 2: “We recruited patients and caregivers from a single VA medical center where the investigators worked; while this approach allowed for home-based interviewed and facilitated participation among individuals with functional impairment, the findings for patients and caregivers may be less representative than those of participants recruited from 6 sites nationally.”

Regarding whether the intervention was co-designed, we did not use a formal co-design process in which end-users were involved in the study planning phase or helped determine the research agenda. However, we did strive to incorporate end-user feedback throughout the intervention design process. Specifically, qualitative interviews focused on barriers and facilitators to measuring functional status in primary care were used to inform the design of the intervention. Staff at each staff also reviewed the process maps of functional measurement for their site. In addition, we completed usability testing and iterative refinement of the electronic screening tool with intended users (licensed vocational nurses and PCPs). The use of qualitative interviews is described throughout the methods; we describe the usability testing process as follows:

Methods, page 10, paragraph 2: “The VA Office of Human Factors Engineering completed Human Factors analysis of the screening tool. We then conducted usability testing and iterative refinement with intended users (licensed vocational nurses [LVNs] and PCPs).”

Responses to Reviewer #2:

1. Reviewer #2: This article effectively utilizes qualitative methods to investigate the implemntation of new measures designed to ensure the full scale of functional status for older veterans is accurately captured and applied in health care settings. Data is well grounded in semi-structured interviews across multiple VA geographical regions. This article effectively utilizes matrix analysis to locate central themes and reveal compelling findings about the screening and assessment of older adults.

a. Response: We thank the reviewer for these comments.

We thank the Editor and Reviewers for their time and efforts to improve our manuscript. 

Sincerely,

Rebecca Brown, MD, MPH

---

## [Editor Report · Decision Letter 1]

15 Aug 2023

Using Multiple Qualitative Methods to Inform Intervention Development: Improving Functional Status Measurement for Older Veterans in Primary Care Settings

PONE-D-23-09223R1

Dear Dr. Brown,

We’re pleased to inform you that your manuscript has been judged scientifically suitable for publication and will be formally accepted for publication once it meets all outstanding technical requirements.

Kind regards,

Mobolanle Balogun

Academic Editor

PLOS ONE
---

## [Editor Report · Acceptance letter]

16 Aug 2023

PONE-D-23-09223R1 

Using Multiple Qualitative Methods to Inform Intervention Development: Improving Functional Status Measurement for Older Veterans in Primary Care Settings 

Dear Dr. Brown:

I'm pleased to inform you that your manuscript has been deemed suitable for publication in PLOS ONE. Congratulations! Your manuscript is now with our production department. 

Kind regards, 

on behalf of

Dr. Mobolanle Balogun 

Academic Editor

PLOS ONE